# Observational study of haloperidol in hospitalized patients with COVID-19

Nicolas Hoertel[1,2,3], Marina Sánchez-Rico[1,4]*, Raphaël Vernet[5], Anne-Sophie Jannot[3,5,6], Antoine Neuraz[6,7], Carlos Blanco[8], Cédric Lemogne[1,2,3], Guillaume Airagnes[1,2,3], Nicolas Paris[9,10], Christel Daniel[9,11], Alexandre Gramfort[12], Guillaume Lemaitre[12], Mélodie Bernaux[13], Ali Bellamine[14], Nathanaël Beeker[14], Frédéric Limosin[1,2,3], on behalf of the AP-HP/Universities/INSERM Covid-19 research collaboration and AP-HP Covid CDR Initiative[¶]

1 Département de Psychiatrie, Hôpital Corentin-Celton, Centre Université de Paris, AP-HP, Issy-les-Moulineaux, France, 2 INSERM, Institut de Psychiatrie et Neurosciences de Paris, Paris, France, 3 Faculté de Santé, UFR de Médecine, Université de Paris, Paris, France, 4 Faculty of Psychology, Department of Psychobiology & Behavioural Sciences Methods, Universidad Complutense de Madrid, Madrid, Spain, 5 Biostatistics and Public Health Department, Hôpital Européen Georges Pompidou, Medical Informatics, AP-HP, Centre-Université de Paris, Paris, France, 6 INSERM, UMR_S 1138, Cordeliers Research Center, Université de Paris, Paris, France, 7 Department of Medical Informatics, Necker-Enfants Malades Hospital, AP-HP, Centre-Université de Paris, Paris, France, 8 National Institute on Drug Abuse, Bethesda, MD, United States of America, 9 AP-HP, DSI-WIND, Paris, France, 10 LIMSI, CNRS, Université Paris-Sud, Université Paris-Saclay, Orsay, France, 11 Sorbonne University, University Paris 13, Sorbonne Paris Cité, INSERM UMR_S 1142, Paris, France, 12 Université Paris-Saclay, Inria, CEA, Palaiseau, France, 13 Direction de la stratégie et de la transformation, AP-HP, Paris, France, 14 Unité de Recherche clinique, Hôpital Cochin, AP-HP, Centre-Université de Paris, Paris, France

¶ Membership of the AP-HP/Universities/INSERM Covid-19 research collaboration and AP-HP Covid CDR Initiative is provided in the Acknowledgments.
* marinals@ucm.es

**Data Availability Statement:** Data can be available at https://eds.aphp.fr// upon request from AP-HP Health Data Warehouse (Entrepôt de Données de Santé (EDS)). The authors did not have special

## Abstract

### Background

Haloperidol, a widely used antipsychotic, has been suggested as potentially useful for patients with COVID-19 on the grounds of its *in-vitro* antiviral effects against SARS-CoV-2, possibly through sigma-1 receptor antagonist effect.

### Methods

We examined the associations of haloperidol use with intubation or death and time to discharge home among adult patients hospitalized for COVID-19 at Assistance Publique-Hôpitaux de Paris (AP-HP) Greater Paris University hospitals. Study baseline was defined as the date of hospital admission. The primary endpoint was a composite of intubation or death and the secondary endpoint was discharge home among survivors in time-to-event analyses. In the primary analyses, we compared these two outcomes between patients receiving and not receiving haloperidol using univariate Cox regression models in matched analytic samples based on patient characteristics and other psychotropic medications. Sensitivity analyses included propensity score analyses with inverse probability weighting and multivariable Cox regression models.

**Funding:** The authors received no specific funding for this work.

**Competing interests:** I have read the journal's policy and the authors of this manuscript have the following competing interests: NH has received personal fees and non-financial support from Lundbeck, outside the submitted work. FL has received speaker and consulting fees from Janssen-Cilag, Euthérapie-Servier, and Lundbeck, outside the submitted work. CL reports personal fees and non-financial support from Janssen-Cilag, Lundbeck, Otsuka Pharmaceutical, and Boehringer Ingelheim, outside the submitted work. GA reports personal fees from Pfizer, Pierre Fabre and Lundbeck, outside the submitted work. Other authors declare no competing interests. This does not alter our adherence to PLOS ONE policies on sharing data and materials.

## Results

Of 15,121 adult inpatients with a positive COVID-19 PT-PCR test, 39 patients (0.03%) received haloperidol within the first 48 hours of admission. Over a mean follow-up of 13.8 days (SD = 17.9), 2,024 patients (13.4%) had a primary end-point event and 10,179 patients (77.6%) were discharged home at the time of study end on May 1$^{st}$. The primary endpoint occurred in 9 patients (23.1%) who received haloperidol and 2,015 patients (13.4%) who did not. The secondary endpoint of discharge home occurred in 16 patients (61.5%) who received haloperidol and 9,907 patients (85.8%) who did not. There were no significant associations between haloperidol use and the primary (HR, 0.80; 95% CI, 0.39 to 1.62, p = 0.531) and secondary (HR, 1.30; 95% CI, 0.74 to 2.28, p = 0.355) endpoints. Results were similar in multiple sensitivity analyses.

## Conclusion

Findings from this multicenter observational study suggest that haloperidol use prescribed at a mean dose of 4.5 mg per day (SD = 5.2) for a mean duration of 8.4 days (SD = 7.2) may not be associated with risk of intubation or death, or with time to discharge home, among adult patients hospitalized for COVID-19.

## Introduction

The novel coronavirus SARS-CoV-2, the causative agent of coronavirus disease 2019 (COVID-19), has caused worldwide health, social and economic disruption [1,2]. In the absence of antiviral medications with proven clinical efficacy [3,4], the search for an effective treatment for patients with COVID-19 among all available medications is urgently needed [4,5].

Based on advances in the knowledge of molecular details of SARS-CoV-2 infection [4], it has been suggested that two sets of pharmacological agents that show in-vitro antiviral activity should be prioritized in that search: the inhibitors of mRNA translation and the predicted regulators of the Sigma1 and Sigma2 receptors [4]. Molecules that target Sigma receptors may reduce virus infectivity through different mechanisms, including lipid remodeling and endoplasmic reticulum stress response [4,6].

Haloperidol, a butyrophenone-derivative antipsychotic widely used in the treatment of psychoses and delirium, has been suggested as potentially useful for patients with COVID-19 on the grounds of its *in-vitro* antiviral effects against SARS-CoV-2, possibly through sigma-1 receptor antagonist effect [4,7].

Short-term use of haloperidol is generally well tolerated [8], although side effects can occur, including extrapyramidal symptoms and QT interval prolongation [9].

To our knowledge, no clinical study has examined to date the potential usefulness of haloperidol in patients hospitalized for COVID-19. Observational studies of patients with COVID-19 taking medications for other indications can help decide which should be prioritized for randomized clinical trials and minimize the risk for patients of being exposed to potentially harmful and ineffective treatments.

To this end, we took advantage of the Assistance Publique-Hôpitaux de Paris (AP-HP) Health Data Warehouse, which includes data on all patients with COVID-19 who had been consecutively admitted to any of the 39 AP-HP Greater Paris University hospitals.

In this report, we examined the associations of haloperidol use with the risk of intubation or death and the time to discharge home among adult patients who have been admitted to AP-HP hospitals for COVID-19. We hypothesized that haloperidol use would be associated with lower risk of a composite endpoint of intubation or death, and with shorter time from hospital admission to discharge home in time-to-event analyses adjusting for patient characteristics and other psychotropic medications.

## Methods

### Setting

We conducted a multicenter observational retrospective study at AP-HP, which includes 39 hospitals, 23 of which are acute, 20 are adult and 3 are pediatric hospitals. We included all adults aged 18 years or over who have been admitted for COVID-19 to these medical centers from the beginning of the epidemic in France, i.e. January 24th, until May 1st. COVID-19 was ascertained by a positive reverse-transcriptase–polymerase-chain-reaction (RT-PCR) test from analysis of nasopharyngeal or oropharyngeal swab specimens. This observational non-interventional retrospective study using routinely collected data received approval from the Institutional Review Board of the AP-HP clinical data warehouse (decision CSE-20-20_COVID19, IRB00011591). AP-HP clinical Data Warehouse initiative ensures patients' information and consent regarding the different approved studies through a transparency portal in accordance with European Regulation on data protection and authorization n˚1980120 from National Commission for Information Technology and Civil Liberties (CNIL). Participants who did not consent to participate in the study were excluded prior to the construction of the database. All procedures related to this work adhered to the ethical standards of the relevant national and institutional committees on human experimentation and with the Helsinki Declaration of 1975, as revised in 2008.

### Data sources

We used data from the AP-HP Health Data Warehouse ('Entrepôt de Données de Santé (EDS)'). This warehouse contains all the clinical data available on all inpatient visits for COVID-19 to any AP-HP hospital. The data obtained included patient demographic characteristics, RT-PCR test results, medication administration data, medication lists during current and past hospitalizations in AP-HP hospitals, current diagnoses, discharge disposition, ventilator use data, and death certificates.

### Variables assessed

We obtained the following data for each patient at the time of the hospitalization: sex; age (binarized at the median value observed in the full sample); hospital, which was categorized into 2 classes following the administrative clustering of AP-HP hospitals in Paris and its suburbs based on their geographical location (i.e., AP-HP Centre–Paris University, Henri Mondor University Hospitals and at home hospitalization; and AP-HP Nord and Hôpitaux Universitaires Paris Seine-Saint-Denis, Paris Saclay University, and Sorbonne University); obesity, defined as having a body-mass index higher than 30 kg/m$^2$ or an International Statistical Classification of Diseases and Related Health Problems (ICD-10) diagnosis code for obesity (E66.0, E66.1, E66.2, E66.8, E66.9); self-reported current smoking status; any medical conditions associated with increased risk of severe COVID-19 [10–14] based on ICD-10 diagnosis codes, including diabetes mellitus (E11), diseases of the circulatory system (I00-I99), diseases of the respiratory system (J00-J99), neoplasms (C00-C96), and diseases of the blood and blood-

forming organs and certain disorders involving the immune mechanism (D5-D8); and any medication prescribed according to compassionate use or as part of a clinical trial (e.g. hydroxychloroquine, azithromycin, remdesivir, tocilizumab, sarilumab, or dexamethasone). To take into account possible confounding by indication bias for haloperidol, we recorded whether patients had any current current diagnosis of psychiatric disorders, including delirium (F00-F99 and R41.0), any prescribed antipsychotic other than haloperidol, and any other prescribed psychotropic medication (i.e., antidepressants, benzodiazepines, Z-drugs, and mood stabilizers, including lithium and antiepileptic medications with mood stabilizing effects).

All medical notes and prescriptions are computerized in Greater Paris University hospitals. Medications including their dose, frequency, date, and mode of administration were identified from medication administration data or scanned hand-written medical prescriptions, through two deep learning models based on BERT contextual embeddings [15], one for the medications and one for their mode of administration. The model was trained on the APmed corpus [16], a previously annotated dataset for this task. Extracted medications names were then normalized to the Anatomical Therapeutic Chemical (ATC) terminology using approximate string matching.

## Haloperidol use

Study baseline was defined as the date of hospital admission. Haloperidol use was defined as receiving haloperidol within the first 48 hours of hospital admission and before the end of the index hospitalization or intubation or death. We used this delay because we considered that, in a context of overwhelming of all hospital units during the COVID-19 peak incidence, patients may not have received or been prescribed the treatment the first day of their admission, or the treatment may not have been recorded in the computerized medication administration data the first day of admission. In this observational study, no specific clinical guidelines were given to practitioners to administer haloperidol.

## Endpoints

The primary endpoint was the time from study baseline to intubation or death. For patients who died after intubation, the timing of the primary endpoint was defined as the time of intubation. The secondary outcome was the time from study baseline to discharge home among survivors. Patients without an end-point event had their data censored on May 1st, 2020.

## Statistical analysis

We calculated frequencies and means (± standard deviations (SD)) of each variable described above in patients receiving or not receiving haloperidol and compared them using standardized mean differences (SMD). A SMD higher than 0.1 was considered to reflect substantial imbalance [17].

To examine the association of haloperidol use with the primary composite endpoint of intubation or death and the secondary endpoint of discharge home among survivors, we performed Cox proportional-hazards regression models [18]. To help account for the nonrandomized prescription of haloperidol and reduce the effects of confounding, the primary analysis used a univariate Cox regression model in a matched analytic sample for each outcome. We selected four controls for each exposed case, based on age, sex, hospital, obesity, smoking status, any medical condition, any medication prescribed according to compassionate use or as part of a clinical trial, any current diagnosis of psychiatric disorders, any prescribed antipsychotic other than haloperidol, and any other prescribed psychotropic medication [19–22]. To reduce the effects of confounding, optimal matching was used in order to obtain the

smallest average absolute distance across all these characteristics between each exposed patient and its corresponding non-exposed matched controls [23]. Weighted Cox regression models were used when proportional hazards assumption was not met [24]. Kaplan-Meier curves were performed [25] and their 95% pointwise confidence intervals were estimated using the nonparametric bootstrap method [26].

We conducted six sensitivity analyses. First, we performed propensity score analyses with inverse probability weighting (IPW) [27,28]. The individual propensities for haloperidol prescription were estimated by a multivariable logistic regression model that included as covariates the same variables used in the primary analyses. The predicted probabilities from the propensity-score model were used to calculate the stabilized inverse-probability-weighting weights [27]. Associations between haloperidol use and the two outcomes were then estimated using IPW Cox regression models. In cases of non-balanced covariates, IPW multivariable Cox regression models adjusting for these non-balanced covariates were also performed. Kaplan-Meier curves were performed using the inverse-probability-weighting weights [25,26]. Second, we performed multivariable Cox regression models including as covariates the same variables as in the primary analyses and the inverse-probability-weighted analyses. Third, to address a potential immortality bias in the exposed group due to a treatment initiation after hospital admission, we performed multivariable cox regression models while considering haloperidol use as a time dependent variable [18], including all participants who received haloperidol at any time from hospital admission until the end of the index hospitalization or intubation or death. In this type of analysis, patients who received haloperidol after study baseline were allowed to come into the analysis risk-sets at the time of actual first initiation of haloperidol. Fourth, we examined whether our findings were similar in models imputing missing data using multiple imputation [29] instead of excluding patients with any missing data as done in the main analyses. Fifth, in order to account for potential latent effects of the variable hospital, we examined whether our findings were similar while considering this variable as a random effects covariate in the main analyses [18]. Finally, because psychotropic medications other than haloperidol could have been prescribed to patients after they received haloperidol, we examined whether the results were similar when not including these variables as covariates in the main analyses.

We also performed additional analyses. First, to increase our confidence that the results might not be due to unmeasured confounding or indication bias, we examined whether the two endpoints differed between patients receiving haloperidol only in the 3 months before hospital admission and those who received it only during the visit. Second, we examined a potential dose-effect relationship by testing the association between the daily dose received (dichotomized at the median value) with the two endpoints among patients who received haloperidol.

For all associations, we performed residual analyses to assess the fit of the data, check assumptions, including proportional hazards assumption using proportional hazards tests and diagnostics based on weighted residuals [30], and examined the potential presence and influence of outliers [31]. Statistical significance was fixed *a priori* at p<0.05. All analyses were conducted in R software version 2.4.3. Statistical code used for the main analyses has been deposited in a recognized public source code repository (GitHub, https://github.com/mlsrico/haloperidol_and_covid19).

## Results

### Characteristics of the cohort

Of the 17,076 hospitalized adult patients with a positive COVID-19 RT-PCR test, 1,908 patients (11.2%) were excluded because of missing data or their young age (i.e. less than 18

years of age). Of the 86 adult patients who received haloperidol at any time during the visit, 47 (54.7%) patients were excluded because they received it more than 48 hours from hospital admission (N = 47, 54.7%) or after intubation (N = 16, 18.6%). Of the remaining 15,121 inpatients, 39 patients (0.3%) received haloperidol at baseline (i.e., within the first 48 hours of hospital admission) at a median daily dose of 3.0 mg per day (SD = 5.2; mean = 4.5; first quartile = 1.0; third quartile = 5.0; range = 0.5–20.0 mg) for a median duration of 7.0 days (SD = 7.2; mean = 8.4; first quartile = 2.5; third quartile = 12.0; range = 1–26). Of these 39 patients, 4 patients (10.2%) had a medication administration by intramuscular injection. Median delay between study baseline and haloperidol initiation was lower than 0 day (SD = 1.0; mean = 0.6; first quartile = 0.0; third quartile = 1.0; range = 0–2 days) (S1 Fig).

First positive COVID-19 RT-PCR tests were obtained after a median delay of 1.2 days (SD = 12.7) from study baseline. This delay was significantly but not substantially different between patients receiving or not receiving haloperidol [median in the exposed group = 1.0 day (SD = 11.2); median in the non-exposed group = 1.2 days (SD = 12.8); Mood's median test Chi-square = 3.76, p = 0.001].

Over a median follow-up of 7 days (SD = 17.9; mean = 13.8; first quartile = 1.0; third quartile = 22.0; range: 1–98 days), 2,024 patients (13.4%) had a primary end-point event and 10,179 patients (77.6%) were discharged home at the time of study end on May 1st. Patients receiving haloperidol had a median follow-up of 7 days (SD = 14.0; mean = 10.9; first quartile = 4; third quartile = 22; range: 1–80 days), while the non-exposed group had a median follow-up of 7 days (SD = 17.9; mean = 13.8; first quartile = 1; third quartile = 22; range: 1–98 days) [Mood's median test Chi-square = 0.41, p = 0.679].

All baseline characteristics, when examined independently, were significantly associated with both endpoints (S1 and S2 Tables). In the multivariable analysis, these associations remained significant for the primary outcome, except for hospital type, smoking status, any medication according to compassionate use or as part of a clinical trial, and any antipsychotic other than haloperidol, as they did for the secondary outcome, except for smoking status.

The distribution of the patient characteristics according to haloperidol use is shown in Table 1. In the full sample, haloperidol use substantially differed (i.e., SMD>0.1) across all characteristics except for any medication according to compassionate use or as part of a clinical trial. After applying the propensity score weights, all these differences became not substantial (i.e., SMD<0.1), except for any antipsychotic other than haloperidol and any other psychotropic medication (Table 1). In the matched analytic sample comprising 195 patients (i.e., 39 patients who received haloperidol at baseline and 156 patients who did not receive haloperidol during the visit from the matched group), there were no substantial differences in patient characteristics according to haloperidol use (all SMD<0.1) (Table 1).

## Study endpoints

The primary endpoint event of intubation or death respectively occurred in 9 patients (23.1%) who received haloperidol and 2,015 patients (13.4%) who did not (Table 2). In both the crude, unadjusted analysis and the primary analysis using a univariate Cox regression model in the matched analytic sample, there were no significant associations between haloperidol use and the primary endpoint (hazard ratio (HR), 1.68; 95% CI, 0.87 to 3.23; p = 0.120; and HR, 0.80; 95% CI, 0.39 to 1.62; p = 0.531, respectively) (Fig 1; Table 2).

Among survivors, the secondary endpoint of discharge home occurred in 16 patients (61.5%) who were prescribed haloperidol and 9,907 patients (85.8%) who were not. Haloperidol use was significantly and negatively associated with the secondary endpoint in the crude, unadjusted analysis (HR, 0.24; 95% CI, 0.13 to 0.44; p<0.001), but this association was not

**Table 1. Characteristics of hospitalized patients with COVID-19 receiving or not receiving haloperidol in the full sample and in the matched analytic sample.**

| | Exposed to haloperidol N = 39 | Not exposed to haloperidol N = 15,082 | Non-exposed matched group N = 156 | Exposed to haloperidol vs. Not exposed to haloperidol (crude analysis) | Exposed to haloperidol vs. Not exposed to haloperidol (analysis weighted by inverse-probability-weighting weights) | Exposed to Haloperidol vs. Non-exposed matched group (crude analysis in the matched analytic sample using a 1:4 ratio) |
|---|---|---|---|---|---|---|
| | N (%) | N (%) | N (%) | SMD | SMD | SMD |
| *Characteristics* | | | | | | |
| Age | | | | 0.592* | 0.015 | 0.031 |
| *18 to 57 years* | 9 (23.1%) | 7,611 (50.5%) | 34 (21.8%) | | | |
| *More than 57 years* | 30 (76.9%) | 7,471 (49.5%) | 122 (78.2%) | | | |
| Sex | | | | 0.182* | 0.054 | 0.039 |
| *Women* | 24 (61.5%) | 7,926 (52.6%) | 93 (59.6%) | | | |
| *Men* | 15 (38.5%) | 7,156 (47.4%) | 63 (40.4%) | | | |
| Hospital | | | | 0.328* | 0.026 | 0.071 |
| *AP-HP Centre–Paris University, Henri Mondor University Hospitals and at home hospitalization* | 12 (30.8%) | 7,017 (46.5%) | 43 (27.6%) | | | |
| *AP-HP Nord and Hôpitaux Universitaires Paris Seine-Saint-Denis, Paris Saclay University and Sorbonne University* | 27 (69.2%) | 8,065 (53.5%) | 113 (72.4%) | | | |
| Smoking | | | | 0.366* | 0.065 | <0.001 |
| *Yes* | 11 (28.2%) | 2,048 (13.6%) | 44 (28.2%) | | | |
| *No* | 28 (71.8%) | 13,034 (86.4%) | 112 (71.8%) | | | |
| Obesity <sup>α</sup> | | | | 0.408* | 0.024 | <0.001 |
| *Yes* | 9 (23.1%) | 1,284 (8.5%) | 36 (23.1%) | | | |
| *No* | 30 (76.9%) | 13,798 (91.5%) | 120 (76.9%) | | | |
| Any medical condition <sup>β</sup> | | | | 0.510* | 0.007 | <0.001 |
| *Yes* | 20 (51.3%) | 4,096 (27.2%) | 80 (51.3%) | | | |
| *No* | 19 (48.7%) | 10,986 (72.8%) | 76 (48.7%) | | | |
| Any medication according to compassionate use or as part of a clinical trial | | | | 0.073 | 0.066 | <0.001 |
| *Yes* | 4 (10.3%) | 1,897 (12.6%) | 16 (10.3%) | | | |
| *No* | 35 (89.7%) | 13,185 (87.4%) | 140 (89.7%) | | | |
| Any current psychiatric disorder or delirium <sup>¥</sup> | | | | 0.842* | 0.080 | <0.001 |
| *Yes* | 15 (38.5%) | 930 (6.2%) | 60 (38.5%) | | | |
| *No* | 24 (61.5%) | 14,152 (93.8%) | 96 (61.5%) | | | |
| Any antipsychotic (other than haloperidol) | | | | 1.154* | 0.240* | <0.001 |
| *Yes* | 18 (46.2%) | 467 (3.10%) | 72 (46.2%) | | | |
| *No* | 21 (53.8%) | 14,615 (96.9%) | 84 (53.8%) | | | |
| Any other psychotropic medication <sup>Ω</sup> | | | | 1.721* | 0.220* | <0.001 |
| *Yes* | 31 (79.5%) | 2169 (14.4%) | 124 (79.5%) | | | |

(*Continued*)

**Table 1.** (Continued)

| | Exposed to haloperidol N = 39 | Not exposed to haloperidol N = 15,082 | Non-exposed matched group N = 156 | Exposed to haloperidol vs. Not exposed to haloperidol (crude analysis) | Exposed to haloperidol vs. Not exposed to haloperidol (analysis weighted by inverse-probability-weighting weights) | Exposed to Haloperidol vs. Non-exposed matched group (crude analysis in the matched analytic sample using a 1:4 ratio) |
|---|---|---|---|---|---|---|
| | N (%) | N (%) | N (%) | SMD | SMD | SMD |
| *No* | 8 (20.5%) | 12,913 (85.6%) | 32 (20.5%) | | | |

[α] Defined as having a body-mass index higher than 30 kg/m$^2$ or based on ICD-10 codes (E66.0, E66.1, E66.2, E66.8, E66.9).

[β] Included diabetes milletus (E11), diseases of the circulatory system (I00-I99), diseases of the respiratory system (J00-J99), neoplasms (C00-C96), and diseases of the blood and blood-forming organs and certain disorders involving the immune mechanism (D5-D8) based on ICD-10 codes.

[¥] Assessed using ICD-10 codes (F00-F99 or R41.0).

[Ω] Included any antidepressant, benzodiazepine, Z-drug, or mood stabilizer (i.e., lithium or antiepileptic medications with mood stabilizing effects).

[*] A SMD higher than 0.1 indicates substantial imbalance.

Abbreviation: SMD, standardized mean difference.

significant in the primary analysis using a univariate Cox regression model in the matched analytic sample (HR, 1.30; 95% CI, 0.74 to 2.28; p = 0.355) (Fig 2; Table 2).

Sensitivity analyses, including multivariable Cox regression models and propensity score analyses with inverse probability weighting in the full sample yielded similar non-significant results for the two endpoints (Figs 1 and 2; Table 2). Similar non-significant results were found in the inverse probability weighting analyses adjusting for the two unbalanced covariates (i.e. any antipsychotic medication other than haloperidol and any other psychotropic medication) (i.e., HR, 1.32; 95% CI, 0.65 to 2.71; p = 0.441 for the primary outcome; and HR,

**Table 2.** Associations between haloperidol use and the endpoints of intubation or death and discharge home among survivors, in the full sample and in the matched analytic sample of patients hospitalized for COVID-19.

| | Intubation or death | Discharge home among survivors |
|---|---|---|
| *Full sample* | | |
| Number of events/Number of patients (%) | 2,024/15,121 (13.4%) | 9,923/11,572 (85.8%) |
| *Haloperidol* | 9/39 (23.1%) | 16/26 (61.5%) |
| *No haloperidol* | 2,015/15,082 (13.4%) | 9,907/11,546 (85.8%) |
| Crude analysis HR (95% CI; p-value) | 1.68 (0.87–3.23; 0.120) | 0.24 (0.13–0.44; <0.001*) |
| Multivariable analysis HR (95% CI; p-value) | 0.55 (0.15–1.98; 0.360) | 0.95 (0.52–1.71; 0.856) |
| Propensity score analysis with inverse probability weighting HR (95% CI; p-value) | 1.31 (0.69–2.49; 0.413) | 1.10 (0.67–1.82; 0.709) |
| *Matched analytic sample* | | |
| Number of events/Number of patients (%) | 59/195 (30.3%) | 68/130 (52.3%) |
| *Haloperidol* | 9/39 (23.1%) | 16/26 (61.5%) |
| *No haloperidol* | 50/156 (32.1%) | 52/104 (50.0%) |
| Crude analysis HR (95% CI; p-value) | 0.80 (0.39–1.62; 0.531) | 1.30 (0.74–2.28; 0.355) |

[*] p-value is significant (p<0.05).

Abbreviations: HR, hazard ratio; CI, confidence interval.

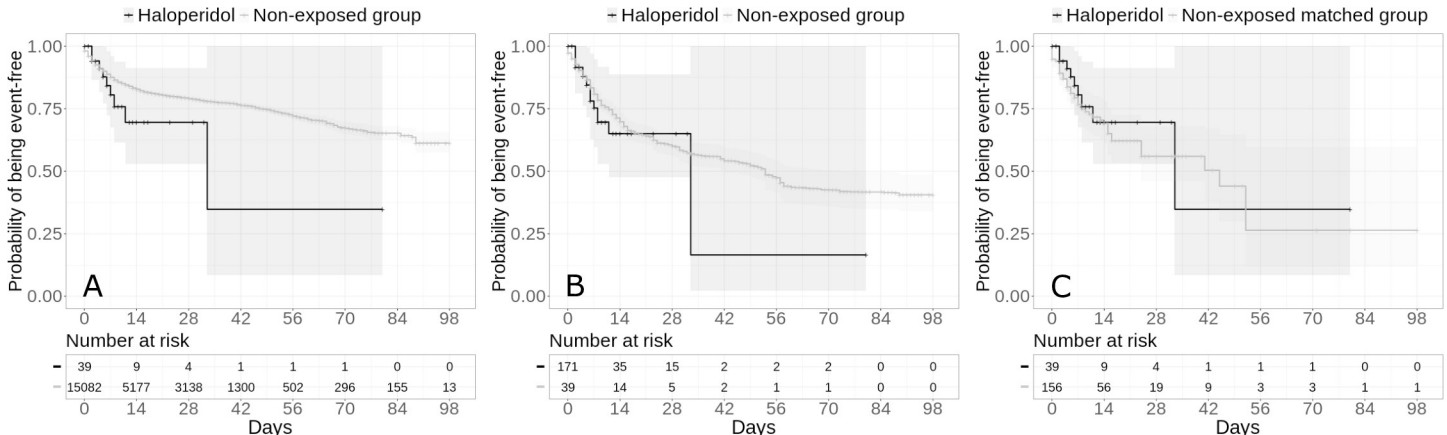

**Fig 1.** Kaplan-Meier curves for intubation or death in the full sample crude analysis (N = 15,121) (A), in the full sample analysis with inverse probability weighting (N = 15,121) (B) and in the matched analytic sample using a 1:4 ratio (N = 195) (C) of patients who had been hospitalized for COVID-19, according to haloperidol use. The shaded areas represent pointwise 95% confidence intervals.

1.18; 95% CI, 0.71 to 1.95; p = 0.528 for the secondary outcome), as well as in the analyses including all participants who received haloperidol during the hospitalization until the end of the index hospitalization or intubation or death and considering haloperidol use as a time dependent variable (S3 Table). Findings were also similar when considering hospital as a random effects variable (S4 Table). Finally, models imputing missing data using multiple imputation yielded very similar results as in the main analyses (S5 Table), as did models not including psychotropic medications other than haloperidol as covariates (S6 Table).

Additional analyses indicated that risks for both endpoints were not significantly different between patients who were prescribed haloperidol only in the three months before the hospitalization and those who received it only during the visit (S7 Table). Exposure to higher rather than lower doses of haloperidol was not significantly associated with the primary or secondary endpoints (S8 and S9 Tables).

A post-hoc analysis indicated that in the full sample, we had 80% power to detect unweighted and unadjusted hazard ratios of at least 0.20/2.41 for the primary endpoint and

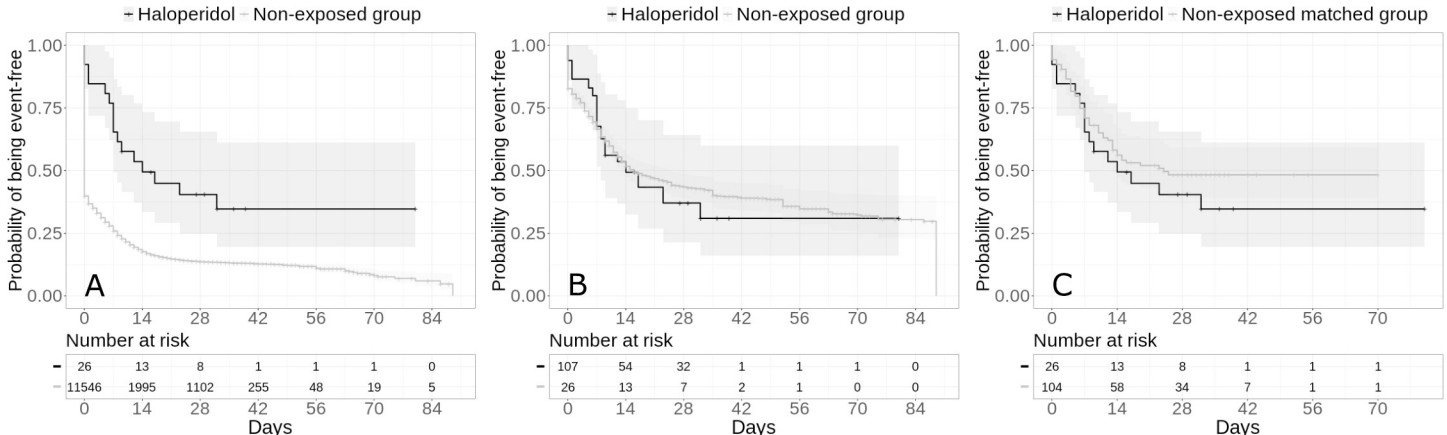

**Fig 2.** Kaplan-Meier curves for discharge home among survivors in the full sample crude analysis (N = 11,572) (A), in the full sample analysis with inverse probability weighting (N = 11,572) (B) and in the matched analytic sample using a 1:4 ratio (N = 130) (C) of patients who had been hospitalized for COVID-19, according to haloperidol use. The shaded areas represent pointwise 95% confidence intervals.

0.27/5.57 for the secondary endpoint, while we had 80% power to detect unweighted and unadjusted hazard ratios of at least 0.39/2.18 for the primary endpoint and 0.39/3.07 for the secondary endpoint in the matched analytic sample.

## Discussion

In this multicenter retrospective observational study involving a large number of adult patients hospitalized for COVID-19, the risk of intubation or death and the time to discharge home among survivors were not significantly different between patients who received haloperidol and those who did not. Although these findings should be interpreted with caution due to the observational design, the relatively wide confidence intervals for estimates, and the fact that this is, to our knowledge, the first study examining these associations in patients with COVID-19, they suggest that haloperidol prescribed at a mean daily dose of 4.5 mg per day (SD = 5.2) for a mean duration of 8.4 days (SD = 7.2) is not associated with risk of intubation or death, or with time to discharge home, among patients hospitalized for COVID-19.

Our study has several limitations. First, there are two possible major inherent biases in observational studies: unmeasured confounding and confounding by indication. We tried to minimize the effects of confounding in several different ways. First, we used a univariate Cox regression models in matched analytic samples and multivariable Cox regression models with inverse probability weighting to minimize the effects of confounding by indication [27,28]. Second, although some amount of unmeasured confounding may remain, our analyses adjusted for numerous potential confounders, including age, sex, hospital, obesity, current smoking status, any medical condition, any medication prescribed according to compassionate use or as part of a clinical trial, any current diagnosis of psychiatric disorders, any prescribed antipsychotic other than haloperidol, and any other psychotropic medication. Finally, the lack of significant associations between the daily dose of haloperidol and the two endpoints also supports our conclusion.

Other limitations include missing data for some variables (i.e., 11.2%) and potential for inaccuracies in the electronic health records, such as the possible lack of documentation of illnesses or medications, or the misidentification of treatment mode of administration (e.g., dose), especially for hand-written medical prescriptions. However, results remained unchanged when using multiple imputation to account for missing data. Second, given the limited number of patients who received haloperidol, our analyses were powered to detect only substantial effect sizes (i.e., 80% power to detect unweighted and unadjusted hazard ratios of at least 0.20/2.41 for the primary endpoint and 0.27/5.57 for the secondary endpoint in the full sample). In addition, the fact that the inverse probability weighting analyses in the full sample were not successful in balancing two covariates between the two groups (i.e. exposed to haloperidol vs. non-exposed) likely has led to reduced statistical power. Third, haloperidol was prescribed at a relatively low dose, i.e., at a mean daily dose of 4.5 mg per day (SD = 5.2), and its antiviral properties might be observable at higher doses. Fourth, associations reported in our study do not imply causal relationships [32]. Finally, despite the multicenter design, our results may not be generalizable to other settings, e.g. outpatients, or regions.

## Conclusion

Findings form this multicenter observational study suggest that haloperidol prescribed at a mean daily dose of 4.5 mg per day (SD = 5.2) for a mean duration of 8.4 days (SD = 7.2) may not be associated with risk of intubation or death, or with time to discharge home, among patients hospitalized for COVID-19.

## Supporting information

**S1 Fig. Study cohort.**
(DOCX)

**S1 Table. Associations of baseline clinical characteristics with the composite endpoint of intubation or death in the cohort of adult patients hospitalized for COVID-19 (N = 15,121).**
(DOCX)

**S2 Table. Associations of baseline clinical characteristics with the endpoint of discharge home in the cohort of adult patients hospitalized for COVID-19 who survived (N = 11,572).**
(DOCX)

**S3 Table. Associations between haloperidol use and the endpoints of intubation or death and discharge home among survivors, in the full sample and in the matched analytic sample of patients hospitalized for COVID-19, while including all patients who received haloperidol at any time during the visit and before the end of the index hospitalization or intubation or death and considering haloperidol use as a time dependent variable.**
(DOCX)

**S4 Table. Associations between haloperidol use and the endpoints of intubation or death and discharge home among survivors, in the full sample and in the matched analytic sample of patients hospitalized for COVID-19, when considering the variable hospital as a random effects variable.**
(DOCX)

**S5 Table. Associations between haloperidol use and the endpoints of intubation or death and discharge home among survivors, in the full sample and in the matched analytic sample of patients hospitalized for COVID-19, following imputation of missing data with multiple imputation.**
(DOCX)

**S6 Table. Associations between haloperidol use and the endpoints of intubation or death and discharge home among survivors, in the full sample and in the matched analytic sample of patients hospitalized for COVID-19, in multivariable models not including psychotropic medications other than haloperidol as covariates.**
(DOCX)

**S7 Table. Comparing haloperidol use only during the three months before hospitalization versus only during the visit for the endpoints of intubation or death and discharge home among survivors.**
(DOCX)

**S8 Table. Association between haloperidol dose and the endpoint of intubation or death.**
(DOCX)

**S9 Table. Association between haloperidol dose and the endpoint of discharge home among survivors.**
(DOCX)

## Acknowledgments

The authors warmly thank the EDS APHP Covid consortium integrating the APHP Health Data Warehouse team as well as all the APHP staff and volunteers who contributed to the implementation of the EDS-Covid database and operating solutions for this database.

Collaborators EDS APHP Covid consortium: Pierre-Yves ANCEL, Alain BAUCHET, Nathanaël BEEKER, Vincent BENOIT, Mélodie BERNAUX, Ali BELLAMINE, Romain BEY, Aurélie BOURMAUD, Stéphane BREANT, Anita BURGUN, Fabrice CARRAT, Charlotte CAUCHETEUX, Julien CHAMP, Sylvie CORMONT, Christel DANIEL, Julien DUBIEL, Catherine DUCLOAS, Loic ESTEVE, Marie FRANK, Nicolas GARCELON, Alexandre GRAMFORT, Nicolas GRIFFON, Olivier GRISEL, Martin GUILBAUD, Claire HASSEN-KHODJA, François HEMERY, Martin HILKA, Anne Sophie JANNOT, Jerome LAMBERT, Richard LAYESE, Judith LEBLANC, Léo LEBOUTER, Guillaume LEMAITRE, Damien LEPROVOST, Ivan LERNER, Kankoe LEVI SALLAH, Aurélien MAIRE, Marie-France MAMZER, Patricia MARTEL, Arthur MENSCH, Thomas MOREAU, Antoine NEURAZ, Nina ORLOVA, Nicolas PARIS, Bastien RANCE, Hélène RAVERA, Antoine ROZES, Elisa SALAMANCA, Arnaud SANDRIN, Patricia SERRE, Xavier TANNIER, Jean-Marc TRELUYER, Damien VAN GYSEL, Gaël VAROQUAUX, Jill Jen VIE, Maxime WACK, Perceval WAJSBURT, Demian WASSERMANN, Eric ZAPLETAL. Membership of the AP-HP/Universities/INSERM Covid-19 research collaboration and AP-HP Covid CDR Initiative is provided at https://www.aphp.fr/, https://u-paris.fr/, and https://www.inserm.fr/.

## Disclaimer

The views and opinions expressed in this report are those of the authors and should not be construed to represent the views of any of the sponsoring organizations, agencies, or the US government.

## Author Contributions

**Conceptualization:** Nicolas Hoertel, Marina Sánchez-Rico, Anne-Sophie Jannot, Nathanaël Beeker, Frédéric Limosin.

**Data curation:** Marina Sánchez-Rico, Raphaël Vernet, Anne-Sophie Jannot, Antoine Neuraz, Nicolas Paris, Christel Daniel, Alexandre Gramfort, Guillaume Lemaitre, Mélodie Bernaux, Nathanaël Beeker.

**Formal analysis:** Nicolas Hoertel, Marina Sánchez-Rico, Raphaël Vernet.

**Methodology:** Nicolas Hoertel, Marina Sánchez-Rico.

**Writing – original draft:** Nicolas Hoertel.

**Writing – review & editing:** Nicolas Hoertel, Marina Sánchez-Rico, Anne-Sophie Jannot, Carlos Blanco, Cédric Lemogne, Guillaume Airagnes, Nicolas Paris, Christel Daniel, Alexandre Gramfort, Guillaume Lemaitre, Mélodie Bernaux, Ali Bellamine, Nathanaël Beeker, Frédéric Limosin.

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
