## [Decision Letter · Decision Letter 0]

18 Nov 2020

PONE-D-20-30929

Observational Study of Haloperidol in Hospitalized Patients with Covid-19

PLOS ONE

Dear Dr. Marina Sánchez,

Thank you for submitting your manuscript to PLOS ONE. After careful consideration, we feel that it has merit but does not fully meet PLOS ONE’s publication criteria as it currently stands. Therefore, we invite you to submit a revised version of the manuscript that addresses the points raised during the review process.

 The reviewers have raised a number of points which we believe major modifications are necessary to improve the manuscript, taking into account the reviewers' remarks. Please consider and address each of the comments raised by the reviewers before resubmitting the manuscript. This letter should not be construed as implying acceptance, as a revised version will be subject to re-review.

We look forward to receiving your revised manuscript.

Kind regards,

Wisit Cheungpasitporn, MD

Academic Editor

PLOS ONE

Journal Requirements:

2. Please provide more information on how matching was performed; and explain in more detail the clinical guidelines followed by the physicians to administer Haloperidol. Moreover, please avoid any causation statement both in the Abstract and the Discussion.

3. One of the noted authors is a group or consortium [AP-HP / Universities / INSERM Covid-19 research collaboration and AP-HP Covid CDR Initiative ]. In addition to naming the author group, please list the individual authors and affiliations within this group in the acknowledgments section of your manuscript. Please also indicate clearly a lead author for this group along with a contact email address.

4.Thank you for stating the following in the Competing Interests section:

[I have read the journal's policy and the authors of this manuscript have the following competing interests: NH has received personal fees and non-financial support from Lundbeck, outside the submitted work. FL has received speaker and consulting fees from Janssen-Cilag, Euthérapie-Servier, and Lundbeck, outside the submitted work. CL reports personal fees and non-financial support from Janssen-Cilag, Lundbeck, Otsuka Pharmaceutical, and Boehringer Ingelheim, outside the submitted work. GA reports personal fees from Pfizer, Pierre Fabre and Lundbeck, outside the submitted work. Other authors declare no competing interests.].

Reviewers' comments:

Reviewer's Responses to Questions

**Comments to the Author**

1. Is the manuscript technically sound, and do the data support the conclusions?

Reviewer #1: Partly

Reviewer #2: Partly

Reviewer #3: Partly

2. Has the statistical analysis been performed appropriately and rigorously? 

Reviewer #1: No

Reviewer #2: No

Reviewer #3: No

3. Have the authors made all data underlying the findings in their manuscript fully available?

Reviewer #1: Yes

Reviewer #2: No

Reviewer #3: Yes

4. Is the manuscript presented in an intelligible fashion and written in standard English?

Reviewer #1: Yes

Reviewer #2: Yes

Reviewer #3: Yes

5. Review Comments to the Author

Reviewer #1: =========

OVERVIEW:

=========

The authors present an analysis of the observational effect of haloperidol usage in persons with COVID-19 in a large multi-hospital AP-HP Data Warehouse. The study included persons admitted for hospitalization between 1 Jan 2020 and 20 May 2020, and stratified them into two cohorts of either haloperidol (exposed) or non-haloperidol (referent) persons. The overall findings suggested that haloperidol use did not impact the risk of two outcomes: 1) time to intubation or death, and 2) time to discharge to home.

MAJOR CONCERNS:

The definition of the exposed/haloperidol cohort is problematic. In the methods, the authors state that the exposed haloperidol cohort was defined by including persons who had a prescription of haloperidol before hospital admission, but also by including persons who had the haloperidol prescription AFTER hospital admission. Indeed, only 37 of the persons (~36%) in the exposed haloperidol cohort of 104 persons were actively taking haloperidol at time of admission (baseline).

Where this becomes problematic is that analyses were started at date of admission (time = 0), but the prescription of haloperidol could have been started significantly later than this baseline time=0. In time-to-event analyses, this leads to something called "immortality bias" - in other words, persons in the haloperidol cohort who were not taking the drug at time = 0 had to survive to at least the time of initiation of haloperidol (they were immortal - not eligible to die - until after they actually started the haloperidol). In brief, time-to-event analytical methods do not perform properly if FUTURE information is used to stratify persons into exposure cohorts at baseline. Unfortunately, this problem is quite difficult to fix in the framework of your study design. A proper study would include only the 37 persons in your study at time = 0 who were actively taking haloperidol at time of admission. Other "exposed" cohort persons would come into the analysis risk-sets at the time of actual first initiation of haloperidol (left staggered Cox model entry). Alternatively, you could think of haloperidol as a time-dependent covariable in your analyses. Persons are in the referent cohort until they receive their haloperidol prescription.

Another major analytical concern is the decision of variables to include in your propensity methods for balancing the haloperidol and referent cohorts at baseline. In particular, the inclusion of diagnoses related to delirium, psych disorders, and antipsychotic/psychotropic medications are problematic. Indeed, as is shown in Table 1, the IPWs did not balance your haloperidol and referent cohorts adequately on those characteristics. In other words, the lack of a finding may be related to persistent differences in your exposed vs. referent comparisons even after weighting. This concern is more appropriately addressed in the matched analysis (balance was achieved on these covariables for matched analyses).

From an epidemiological standpoint, the matched design results are far more compelling than the complete but weighted inclusion of all persons. Of particular concern is the lack of common support in the propensity modeling (e.g., the particularly large differences in age, and in DX related to delirium and psych conditions). The matched methodology ensures that you are making valid comparisons between the exposed haloperidol and referent persons - as shown in Table 1 the matched referent persons are much more similar to the exposed persons than were even the IPW balanced groups.

Despite the post hoc analysis suggesting that power was better for detecting small HRs for the weighted analysis, the fact that the IPWs were not successful in balancing the two groups (exposed vs. referent) likely reduced power substantially. An alternate methodology could be to only include those persons in the referent cohort who had common support with the exposed persons among propensity scores. Referent persons with covariable vectors completely dissimilar from the exposed group do not help to power a study by inclusion, but are effectively weighted to zero in analyses (i.e., power is much decreased from that estimated from an effective sample size of ~12,000 referent persons). To salvage the current study may require deeper thought about which referent persons and which covariables to include in propensity models to achieve similarity between the exposed and referent persons.

In summary, the study suffers from several major methodological concerns. Chief among them those noted above. In addition, the sample size of the exposed haloperidol cohort is very small. Even if all haloperidol exposed persons are properly included from the time of prescription (not from time of admission), only 104 exposed persons are identified. The authors have salvaged degrees of freedom in Cox models by using IPW methods; however, with only 27 and 26 outcomes in the respective two endpoint analyses, the lack of any significant findings is not surprising. Methodologically, the matched analyses are more compelling and better balanced at baseline than are the IPW weighted analyses, and authors may consider making the matched analysis primary. Nonetheless, while the size of the effect in the matched analyses is suggestively promising (HR = 0.76 for death), the small cohort size and corresponding small number of outcomes hinders more precise estimation.

OTHER CONCERNS:

FIGURE 1 is not very useful. It does not really display any detailed information that is not already reported in a couple sentences in the Results section. If you wish to include this figure, it should include more detailed information about the types of data missing from excluded persons - or some other novel information that is not already reported. Flowcharts with 3 boxes are generally not very compelling from a printed space to information conveyed ratio standpoint.

Figures 2 and 3 should include information below the x-axis of the figure noting the number of persons remaining at risk for each of the two lines displayed. For example at 25, 50, 75, 100, etc. days after baseline – note the number of remaining persons in the haloperidol (exposed) group and in the referent group. These numbers are important for the interpretation of the comparison between haloperidol and referent persons.

In general, time to event durations should be reported as median, quartile 1 (25th percentile), quartile 3 (75th percentile), minimum, and maximum. The distributions of time-to-event durations are almost always heavily right skewed, and means and standard deviations are misleading metrics for reporting.

Reviewer #2: The manuscript presents very important and timely results. This review focuses instead on statistical content.

The results presented in the manuscript may well prove to be accurate and sound. The statistical methods are generally appropriate, but their rigor does not equal the potential importance of this work. The manuscript would be appropriately strengthened by addressing the following deficiencies:

1. The results are based on complete-case analyses and 5% of patient records were excluded due to missing values. That fraction of missing observations is near the threshold of levels which risk bias. The authors reported using a completed-case analysis based on multiple imputation, but did not present that information. That information should be provided at least as a supporting information file, and should be considered to replace the complete-case analysis presented in the manuscript.

2. The 39 hospitals were the primary sampling units, and therefore intra-hospital correlation is another potential source of bias in the estimate of standard errors in the time-to-event analysis. It is reasonable to expect that care of Covid-19 patients might differ among hospitals. For example, hospitals may have variable strengths and weaknesses. Therefore a more convincing analysis would include random effects (frailties) to capture latent hospital effects.

3. Please provide a more detailed description of the methods used to detect violations of the proportional hazards assumption and the construction of the weights used to address that.

4. The analyses are inadequately documented. Methodological statements such as the sentence on line 205 of the manuscript fail to convey any useful or relevant information. The authors should instead provide their code as part of supplemental material.

5. The authors provide a link to the data on line 362. That link points to a general French-language website rather than to specific data. Most readers will have great difficulty navigating through that website to find the data. I could not. Instead, the data file and metadata should be posted in a publicly accessible archive or provided as supporting information.

Reviewer #3: The Authors examined the treatment effect of haloperidol use in hospitalized COVID-19 patients at the APHP Greater Paris University hospitals. The primary endpoint was a composite of intubation or death and the secondary endpoint was discharge to home among survivors in time-to-event analyses. To control selection bias due to nonrandom assignment of treatments, the authors built a propensity score model with demographic and clinical factors and medication uses as predictors for haloperidol use. The authors then calculated stabilized weights and applied the weights to a multivariable Cox regression model with the start time being the date of hospital admission. Sensitivity analyses were also conducted including a multivariable Cox regression model with the same variables as in the propensity score model and a univariate Cox regression model in a matched analytic sample. The manuscript was well organized and clearly written. My major concerns are:

1. For those who received the treatment not on admission date but during hospitalization, their follow-up periods consisted of two parts: untreated and treated. The authors did not consider these two different follow-up periods in their analyses.

2. Related to comment 1, immortal bias may arise because patients had to be alive before receiving the treatment compared to the control group who could die any time after hospital admission.

3. It does not seem right to include medications prescribed during hospitalization in propensity score model. This is because these medications could be given after patients received haloperidol treatment.

4. Table 1 shows that after applying weighting using IPW, several variables were still significantly different between the two groups. I was wondering if the authors explored higher orders and interactions between covariates to improve the propensity score model. Another way is to include these unbalanced covariates in the Cox model.

5. In line 173, the authors mentioned ‘weighted Cox regression models were used when proportional hazards were not met’. I am not aware of such approach for handling the nonproportional hazard issue. Please elaborate this approach and cite the reference(s).

6. I recommend including those who did not consent in Figure 1.

7. I am not sure if the linear form of the dosage is appropriate in the Cox model. I was wondering if the authors had tried different functional forms for this variable when studying the dosage effect.

8. The power estimation did not mean much here because weighting was not considered.

6. PLOS authors have the option to publish the peer review history of their article (what does this mean?). If published, this will include your full peer review and any attached files.

Reviewer #1: **Yes: **Brandon R Grossardt, MS

Reviewer #2: No

Reviewer #3: No

---

## [Author Response · Author response to Decision Letter 0]

17 Dec 2020

Editor

Answer: We carefully complied with PLOS ONE's style requirements.

2. Please provide more information on how matching was performed; and explain in more detail the clinical guidelines followed by the physicians to administer Haloperidol. Moreover, please avoid any causation statement both in the Abstract and the Discussion.

Answer: We now better detail how matching was performed. In addition, we now provide the statistical code used for the main analyses, which has been deposited in a recognized public source code repository (GitHub, https://github.com/mlsrico/haloperidol_and_covid19). In this observational retrospective study, no specific clinical guidelines were given to practitioners to administer haloperidol. Observational studies of patients with COVID-19 taking medications for other indications can help decide which should be prioritized for randomized clinical trials and minimize the risk for patients of being exposed to potentially harmful and ineffective treatments. In the revised version of the article, we also detailed the parameters of haloperidol administration. Finally, we carefully reviewed the writing to avoid any causation statement both in the Abstract and the Discussion.

Following this comment, we modified the manuscript as follows:

P8: “In this observational study, no specific clinical guidelines were given to practitioners to administer haloperidol”

P8: “To help account for the nonrandomized prescription of haloperidol and reduce the effects of confounding, the primary analysis used a univariate Cox regression model in a matched analytic sample for each outcome. We selected four controls for each exposed case, based on age, sex, hospital, obesity, smoking status, any medical condition, any medication prescribed according to compassionate use or as part of a clinical trial, any current diagnosis of psychiatric disorders, any prescribed antipsychotic other than haloperidol, and any other prescribed psychotropic medication [19–21]. To reduce the effects of confounding, optimal matching was used in order to obtain the smallest average absolute distance across all these characteristics between each exposed patient and its corresponding non-exposed matched controls [22].” 

P10: “All analyses were conducted in R software version 2.4.3. Statistical code used for the main analyses has been deposited in a recognized public source code repository (GitHub, https://github.com/mlsrico/haloperidol_and_covid19).”

P11: “Of the remaining 15,121 inpatients, 39 patients (0.3%) received haloperidol at baseline (i.e., within the first 48 hours of hospital admission) at a mean daily dose of 4.5 mg per day (SD=5.2; median=3.0; first quartile=1.0; third quartile=5.0; range=0.5-20.0 mg) for a mean duration of 8.4 days (SD=7.2; median=7.0; first quartile=2.5; third quartile=12.0; range=1-26). Of these 39 patients, 4 patients (10.2%) had a medication administration by intramuscular injection. Mean delay between study baseline and haloperidol initiation was 0.6 days (SD=1.0; median=0.0; first quartile=0.0; third quartile=1.0; range=0-2 days) (S1 Figure).”

P20: “Fourth, associations reported in our study do not imply causal relationships [31]”. 

3. One of the noted authors is a group or consortium [AP-HP / Universities / INSERM Covid-19 research collaboration and AP-HP Covid CDR Initiative]. In addition to naming the author group, please list the individual authors and affiliations within this group in the acknowledgments section of your manuscript. Please also indicate clearly a lead author for this group along with a contact email address.

Answer: Our institution (AP-HP) has decided that all studies on COVID-19 would be published on behalf of a large consortium (i.e. [AP-HP / Universities / INSERM Covid-19 research collaboration and AP-HP Covid CDR Initiative]), which comprises hundreds of workers, included nurses, practitioners, researchers, and administrative staff members. Therefore, we are not able to list hundreds of names and affiliations. We only kept the mention to this consortium and added a lead author for this consortium (i.e., Nicolas Hoertel) and his email address.

[I have read the journal's policy and the authors of this manuscript have the following competing interests: NH has received personal fees and non-financial support from Lundbeck, outside the submitted work. FL has received speaker and consulting fees from Janssen-Cilag, Euthérapie-Servier, and Lundbeck, outside the submitted work. CL reports personal fees and non-financial support from Janssen-Cilag, Lundbeck, Otsuka Pharmaceutical, and Boehringer Ingelheim, outside the submitted work. GA reports personal fees from Pfizer, Pierre Fabre and Lundbeck, outside the submitted work. Other authors declare no competing interests.].

Answer: We added the following sentence to the Competing Interests statement: “This does not alter our adherence to PLOS ONE policies on sharing data and materials. The data that support the findings of this study are available from the AP-HP Health Data Warehouse (Entrepôt de Données de Santé (EDS)). Restrictions apply to the availability of these data, which were used under license for this study. Data are available at https://eds.aphp.fr// with the permission of AP-HP Health Data Warehouse (Entrepôt de Données de Santé (EDS)).” The Competing Interests statement has been added to the cover letter as follows:

Competing Interests Statement

“We have read the journal's policy and the authors of this manuscript have the following competing interests: NH has received personal fees and non-financial support from Lundbeck, outside the submitted work. FL has received speaker and consulting fees from Janssen-Cilag, Euthérapie-Servier, and Lundbeck, outside the submitted work. CL reports personal fees and non-financial support from Janssen-Cilag, Lundbeck, Otsuka Pharmaceutical, and Boehringer Ingelheim, outside the submitted work. GA reports personal fees from Pfizer, Pierre Fabre and Lundbeck, outside the submitted work. Other authors declare no competing interests. This does not alter our adherence to PLOS ONE policies on sharing data and materials. The data that support the findings of this study are available from the AP-HP Health Data Warehouse (Entrepôt de Données de Santé (EDS)). Restrictions apply to the availability of these data, which were used under license for this study. Data are available at https://eds.aphp.fr// with the permission of AP-HP Health Data Warehouse (Entrepôt de Données de Santé (EDS)).”

Reviewer #1

5/ OVERVIEW:

The authors present an analysis of the observational effect of haloperidol usage in persons with COVID-19 in a large multi-hospital AP-HP Data Warehouse. The study included persons admitted for hospitalization between 1 Jan 2020 and 20 May 2020, and stratified them into two cohorts of either haloperidol (exposed) or non-haloperidol (referent) persons. The overall findings suggested that haloperidol use did not impact the risk of two outcomes: 1) time to intubation or death, and 2) time to discharge to home.

MAJOR CONCERNS:

The definition of the exposed/haloperidol cohort is problematic. In the methods, the authors state that the exposed haloperidol cohort was defined by including persons who had a prescription of haloperidol before hospital admission, but also by including persons who had the haloperidol prescription AFTER hospital admission. Indeed, only 37 of the persons (~36%) in the exposed haloperidol cohort of 104 persons were actively taking haloperidol at time of admission (baseline).

Where this becomes problematic is that analyses were started at date of admission (time = 0), but the prescription of haloperidol could have been started significantly later than this baseline time=0. In time-to-event analyses, this leads to something called "immortality bias" - in other words, persons in the haloperidol cohort who were not taking the drug at time = 0 had to survive to at least the time of initiation of haloperidol (they were immortal - not eligible to die - until after they actually started the haloperidol). In brief, time-to-event analytical methods do not perform properly if FUTURE information is used to stratify persons into exposure cohorts at baseline. Unfortunately, this problem is quite difficult to fix in the framework of your study design. A proper study would include only the 37 persons in your study at time = 0 who were actively taking haloperidol at time of admission. Other "exposed" cohort persons would come into the analysis risk-sets at the time of actual first initiation of haloperidol (left staggered Cox model entry). Alternatively, you could think of haloperidol as a time-dependent covariable in your analyses. Persons are in the referent cohort until they receive their haloperidol prescription.

Answer: We thank very much the reviewer for pointing out this potential bias and for indicating how to address it. We followed the reviewer’s suggestion and have rerun all statistical analyses while considering a new definition of haloperidol use to only include patients actively taking haloperidol at study baseline, in order to reduce the risk of immortality bias. Haloperidol use is now defined as receiving haloperidol within the first 48 hours of hospital admission and before the end of the index hospitalization or intubation or death. We used this delay because we considered that, in a context of overwhelming of all hospital units during the COVID-19 peak incidence, patients may not have received or been prescribed their usual treatment the first day of their admission, or the treatment may not have been recorded in the computerized medication administration data the first day of admission. We chose to exclude from the main analyses patients who were prescribed haloperidol more than 48 hours from hospital admission or after intubation, in order to reduce the risk of immortality bias, help interpret results, and because adjustment variables were measured at the time of hospital admission. Furthermore, we followed the reviewer’s suggestion and also performed additional sensitivity analyses considering haloperidol use as a time dependent variable, including all participants who received haloperidol during the hospitalization from study baseline until the end of the index hospitalization or intubation or death, in order to account for untreated and treated periods among patients receiving haloperidol. Results were similar, supporting the robustness of our findings.

Following this comment, we redid all Figures and Tables and modified the manuscript as follows:

P7: “Study baseline was defined as the date of hospital admission. Haloperidol use was defined as receiving haloperidol within the first 48 hours of hospital admission and before the end of the index hospitalization or intubation or death. We used this delay because we considered that, in a context of overwhelming of all hospital units during the COVID-19 peak incidence, patients may not have received or been prescribed the treatment the first day of their admission, or the treatment may not have been recorded in the computerized medication administration data the first day of admission.”

P9: “Third, to address a potential immortality bias in the exposed group due to a treatment initiation after hospital admission, we performed multivariable cox regression models while considering haloperidol use as a time dependent variable [18], including all participants who received haloperidol at any time from hospital admission until the end of the index hospitalization or intubation or death. In this type of analysis, patients who received haloperidol after study baseline were allowed to come into the analysis risk-sets at the time of actual first initiation of haloperidol.”

 P10: “Of the 17,076 hospitalized adult patients with a positive COVID-19 RT-PCR test, 1,908 patients (11.2%) were excluded because of missing data or their young age (i.e. less than 18 years of age). Of the 86 adult patients who received haloperidol at any time during the visit, 47 (54.7%) patients were excluded because they received it more than 48 hours from hospital admission (N=47, 54.7%) or after intubation (N=16, 18.6%). Of the remaining 15,121 inpatients, 39 patients (0.3%) received haloperidol at baseline (i.e., within the first 48 hours of hospital admission) at a mean daily dose of 4.5 mg per day (SD=5.2; median=3.0; first quartile=1.0; third quartile=5.0; range=0.5-20.0 mg) for a mean duration of 8.4 days (SD=7.2; median=7.0; first quartile=2.5; third quartile=12.0; range=1-26). Of these 39 patients, 4 patients (10.2%) had a medication administration by intramuscular injection. Mean delay between study baseline and haloperidol initiation was 0.6 days (SD=1.0; median=0.0; first quartile=0.0; third quartile=1.0; range=0-2 days) (S1 Figure).”

P17-18: “Similar non-significant results were found in the inverse probability weighting analyses adjusting for the two unbalanced covariates (i.e. any antipsychotic medication other than haloperidol and any other psychotropic medication) (i.e., HR, 1.32; 95% CI, 0.65 to 2.71; p=0.441 for the primary outcome; and HR, 1.18; 95% CI, 0.71 to 1.95; p=0.528 for the secondary outcome), as well as in the analyses including all participants who received haloperidol during the hospitalization until the end of the index hospitalization or intubation or death and considering haloperidol use as a time dependent variable (S3 Table).”

6/ Another major analytical concern is the decision of variables to include in your propensity methods for balancing the haloperidol and referent cohorts at baseline. In particular, the inclusion of diagnoses related to delirium, psych disorders, and antipsychotic/psychotropic medications are problematic. Indeed, as is shown in Table 1, the IPWs did not balance your haloperidol and referent cohorts adequately on those characteristics. In other words, the lack of a finding may be related to persistent differences in your exposed vs. referent comparisons even after weighting. This concern is more appropriately addressed in the matched analysis (balance was achieved on these covariables for matched analyses).

From an epidemiological standpoint, the matched design results are far more compelling than the complete but weighted inclusion of all persons. Of particular concern is the lack of common support in the propensity modeling (e.g., the particularly large differences in age, and in DX related to delirium and psych conditions). The matched methodology ensures that you are making valid comparisons between the exposed haloperidol and referent persons - as shown in Table 1 the matched referent persons are much more similar to the exposed persons than were even the IPW balanced groups.

Answer: We thank the reviewer for this comment. Following it, we now consider the matched analyses, for which covariates were adequately balanced, as primary analyses, and the IPW analyses as sensitivity analyses. We also carefully reconsidered the covariates included and their dichotomization in order to reduce covariate imbalance in the IPW analyses. In the matched analytic sample, there were no differences in patient characteristics according to haloperidol use (all standardized mean differences lower than 0.1). However, because two covariates (i.e. any antipsychotic medication other than haloperidol and any other psychotropic medication) were unbalanced in the IPW analysis conducted in the full sample, we also performed an IPW multivariable Cox regression model adjusting for these two non-balanced covariates to examine the robustness of our findings.

Following this comment, we modified the manuscript as follows:

P8: “We calculated frequencies and means (± standard deviations (SD)) of each variable described above in patients receiving or not receiving haloperidol and compared them using standardized mean differences (SMD). A SMD higher than 0.1 was considered to reflect substantial imbalance [17].”

P8: “To help account for the nonrandomized prescription of haloperidol and reduce the effects of confounding, the primary analysis used a univariate Cox regression model in a matched analytic sample for each outcome.”

P9: “Associations between haloperidol use and the two outcomes were then estimated using IPW Cox regression models. In cases of non-balanced covariates, IPW multivariable Cox regression models adjusting for these non-balanced covariates were also performed.”

P12: “The distribution of the patient characteristics according to haloperidol use is shown in Table 1. In the full sample, haloperidol use substantially differed (i.e., SMD>0.1) across all characteristics except for any medication according to compassionate use or as part of a clinical trial. After applying the propensity score weights, all these differences became not substantial (i.e., SMD<0.1), except for any antipsychotic other than haloperidol and any other psychotropic medication (Table 1). In the matched analytic sample comprising 195 patients (i.e., 39 patients who received haloperidol at baseline and 156 patients who did not receive haloperidol during the visit from the matched group), there were no substantial differences in patient characteristics according to haloperidol use (all SMD<0.1) (Table 1).”

P17:“Similar non-significant results were found in the inverse probability weighting analyses adjusting for the two unbalanced covariates (i.e. any antipsychotic medication other than haloperidol and any other psychotropic medication) (i.e., HR, 1.32; 95% CI, 0.65 to 2.71; p=0.441 for the primary outcome; and HR, 1.18; 95% CI, 0.71 to 1.95; p=0.528 for the secondary outcome), as well as in the analyses including all participants who received haloperidol during the hospitalization until the end of the index hospitalization or intubation or death and considering haloperidol use as a time dependent variable (S3 Table).”

7/ Despite the post hoc analysis suggesting that power was better for detecting small HRs for the weighted analysis, the fact that the IPWs were not successful in balancing the two groups (exposed vs. referent) likely reduced power substantially. An alternate methodology could be to only include those persons in the referent cohort who had common support with the exposed persons among propensity scores. Referent persons with covariable vectors completely dissimilar from the exposed group do not help to power a study by inclusion, but are effectively weighted to zero in analyses (i.e., power is much decreased from that estimated from an effective sample size of ~12,000 referent persons). To salvage the current study may require deeper thought about which referent persons and which covariables to include in propensity models to achieve similarity between the exposed and referent persons.

Answer: We agree with the Reviewer and now better acknowledge this limitation related to limited statistical power in the discussion section. We also carefully reconsidered the covariates included and their dichotomization in order to reduce covariate imbalance in the IPW analyses. In the matched analytic sample, there were no differences in patient characteristics according to haloperidol use (all standardized mean differences lower than 0.1). However, because two covariates (i.e. any antipsychotic medication other than haloperidol and any other psychotropic medication) were unbalanced in the IPW analysis conducted in the full sample, we also performed an IPW multivariable Cox regression model adjusting for these two non-balanced covariates to examine the robustness of our findings. Because each analysis has its own limitations, we performed matched analyses (as primary analyses), for which covariates were adequately balanced, and two sensitivity analyses, a propensity score analysis with inverse probability weighting and a multivariable Cox regression model. We consider that the similarity of results across these analytic approaches could support the robustness of our results.

Following this comment, we modified the manuscript as follows (please see also responses to comment #6):

 P8: “To examine the association of haloperidol use with the primary composite endpoint of intubation or death and the secondary endpoint of discharge home among survivors, we performed Cox proportional-hazards regression models [18]. To help account for the nonrandomized prescription of haloperidol and reduce the effects of confounding, the primary analysis used a univariate Cox regression model in a matched analytic sample for each outcome. We selected four controls for each exposed case, based on age, sex, hospital, obesity, smoking status, any medical condition, any medication prescribed according to compassionate use or as part of a clinical trial, any current diagnosis of psychiatric disorders, any prescribed antipsychotic other than haloperidol, and any other prescribed psychotropic medication [19–21]. To reduce the effects of confounding, optimal matching was used in order to obtain the smallest average absolute distance across all these characteristics between each exposed patient and its corresponding non-exposed matched controls [22]. Weighted Cox regression models were used when proportional hazards assumption was not met [23]. Kaplan-Meier curves were performed [24] and their 95% pointwise confidence intervals were estimated using the nonparametric bootstrap method [25].”

P9: “We conducted six sensitivity analyses. First, we performed propensity score analyses with inverse probability weighting (IPW) [26,27]. The individual propensities for haloperidol prescription were estimated by a multivariable logistic regression model that included as covariates the same variables used in the primary analyses. The predicted probabilities from the propensity-score model were used to calculate the stabilized inverse-probability-weighting weights [26]. Associations between haloperidol use and the two outcomes were then estimated using IPW Cox regression models. In cases of non-balanced covariates, IPW multivariable Cox regression models adjusting for these non-balanced covariates were also performed. Kaplan-Meier curves were performed using the inverse-probability-weighting weights [24,25]. Second, we performed multivariable Cox regression models including as covariates the same variables as in the primary analyses and the inverse-probability-weighted analyses. Third, to address a potential immortality bias in the exposed group due to a treatment initiation after hospital admission, we performed multivariable cox regression models while considering haloperidol use as a time dependent variable [18], including all participants who received haloperidol at any time from hospital admission until the end of the index hospitalization or intubation or death. In this type of analysis, patients who received haloperidol after study baseline were allowed to come into the analysis risk-sets at the time of actual first initiation of haloperidol. Fourth, we examined whether our findings were similar in models imputing missing data using multiple imputation [28] instead of excluding patients with any missing data as done in the main analyses. Fifth, in order to account for potential latent effects of the variable hospital, we examined whether our findings were similar while considering this variable as a random effects covariate in the main analyses [18].”

P18: “A post-hoc analysis indicated that in the full sample, we had 80% power to detect unweighted and unadjusted hazard ratios of at least 0.20 / 2.41 for the primary endpoint and 0.27 / 5.57 for the secondary endpoint, while we had 80% power to detect unweighted and unadjusted hazard ratios of at least 0.39 / 2.18 for the primary endpoint and 0.39 / 3.07 for the secondary endpoint in the matched analytic sample.”

P19: “Second, given the limited number of patients who received haloperidol, our analyses were powered to detect only substantial effect sizes (i.e., 80% power to detect unweighted and unadjusted hazard ratios of at least 0.20 / 2.41 for the primary endpoint and 0.27 / 5.57 for the secondary endpoint in the full sample). In addition, the fact that the inverse probability weighting analyses in the full sample were not successful in balancing two covariates between the two groups (i.e. exposed to haloperidol vs. non-exposed) likely has led to reduced statistical power.”

8/ In summary, the study suffers from several major methodological concerns. Chief among them those noted above. In addition, the sample size of the exposed haloperidol cohort is very small. Even if all haloperidol exposed persons are properly included from the time of prescription (not from time of admission), only 104 exposed persons are identified. The authors have salvaged degrees of freedom in Cox models by using IPW methods; however, with only 27 and 26 outcomes in the respective two endpoint analyses, the lack of any significant findings is not surprising. Methodologically, the matched analyses are more compelling and better balanced at baseline than are the IPW weighted analyses, and authors may consider making the matched analysis primary. Nonetheless, while the size of the effect in the matched analyses is suggestively promising (HR = 0.76 for death), the small cohort size and corresponding small number of outcomes hinders more precise estimation.

Answer: Following this comment and comments #5 to #7, we have considered the matched analyses, for which covariates were adequately balanced, as primary analyses, and the IPW analyses and multivariable Cox regression models as sensitivity analyses. In addition, we performed additional sensitivity analyses, i.e. multivariable cox regression models while considering haloperidol use as a time dependent variable, including all participants who received haloperidol at any time from hospital admission until the end of the index hospitalization or intubation or death. In that analysis, patients who received haloperidol after study baseline were allowed to come into the analysis risk-sets at the time of actual first initiation of haloperidol. These sensitivity analyses aimed at evaluating whether our results were robust across different analytic approaches. 

We now consider the matched analyses, for which covariates were adequately balanced, as primary analyses, and the IPW analyses as sensitivity analyses. We also carefully reconsidered the covariates included and their dichotomization in order to reduce covariate imbalance in the IPW analyses, and performed in addition IPW multivariable Cox regression models adjusting for these non-balanced covariates.

Finally, we better acknowledge the limitation related to limited statistical power in the discussion section.

Following this comment, we modified the manuscript. Please see responses to comments #5 to #7.

9/ OTHER CONCERNS:

FIGURE 1 is not very useful. It does not really display any detailed information that is not already reported in a couple sentences in the Results section. If you wish to include this figure, it should include more detailed information about the types of data missing from excluded persons - or some other novel information that is not already reported. Flowcharts with 3 boxes are generally not very compelling from a printed space to information conveyed ratio standpoint.

Answer: Following this comment, we have detailed information about the types of data missing from excluded patients in this Figure, and have moved it to Supplementary material (S1 Figure).

10/ Figures 2 and 3 should include information below the x-axis of the figure noting the number of persons remaining at risk for each of the two lines displayed. For example at 25, 50, 75, 100, etc. days after baseline – note the number of remaining persons in the haloperidol (exposed) group and in the referent group. These numbers are important for the interpretation of the comparison between haloperidol and referent persons.

Answer: Following this comment, we have added the number of persons remaining at risk for each of the two lines displayed in these Figures.

11/ In general, time to event durations should be reported as median, quartile 1 (25th percentile), quartile 3 (75th percentile), minimum, and maximum. The distributions of time-to-event durations are almost always heavily right skewed, and means and standard deviations are misleading metrics for reporting.

Answer: We have added median, quartile 1 (25th percentile), quartile 3 (75th percentile), and range for all time-to-event durations. 

Reviewer #2

12/ The manuscript presents very important and timely results. This review focuses instead on statistical content. The results presented in the manuscript may well prove to be accurate and sound. The statistical methods are generally appropriate, but their rigor does not equal the potential importance of this work.

Answer: We thank the reviewer for this positive comment.

13/ The manuscript would be appropriately strengthened by addressing the following deficiencies: 1. The results are based on complete-case analyses and 5% of patient records were excluded due to missing values. That fraction of missing observations is near the threshold of levels which risk bias. The authors reported using a completed-case analysis based on multiple imputation, but did not present that information. That information should be provided at least as a supporting information file, and should be considered to replace the complete-case analysis presented in the manuscript.

Answer: Following this comment, we now present results based on a complete-case analysis following the exclusion of individuals with any missing data (main analyses) as well as those based on a completed-case analysis based on multiple imputation (supplementary analyses), in order to examine the potential impact of missing data on the results. Results were very similar. 

Following this comment, we have added the S5 Table and modified the manuscript as follows:

P9: “Fourth, we examined whether our findings were similar in models imputing missing data using multiple imputation [28] instead of excluding patients with any missing data as done in the main analyses.”

P18: “Finally, models imputing missing data using multiple imputation yielded very similar results as in the main analyses (S5 Table).”

P19: “Other limitations include missing data for some variables (i.e., 11.2%) and potential for inaccuracies in the electronic health records, such as the possible lack of documentation of illnesses or medications, or the misidentification of treatment mode of administration (e.g., dose), especially for hand-written medical prescriptions. However, results remained unchanged when using multiple imputation to account for missing data.”

14/ The 39 hospitals were the primary sampling units, and therefore intra-hospital correlation is another potential source of bias in the estimate of standard errors in the time-to-event analysis. It is reasonable to expect that care of Covid-19 patients might differ among hospitals. For example, hospitals may have variable strengths and weaknesses. Therefore a more convincing analysis would include random effects (frailties) to capture latent hospital effects.

Answer: Following this comment, we have conducted sensitivity analyses while considering the variable hospital as a random effects variable to capture potential latent hospital effects. Results were very similar. 

Following this comment, we have added a table (S4 Table) and modified the manuscript as follows:

P10: “Fifth, in order to account for potential latent effects of the variable hospital, we examined whether our findings were similar while considering this variable as a random effects covariate in the main analyses [18].”

P18: “Findings were also similar when considering hospital as a random effects variable (S4 Table).” 

15/ Please provide a more detailed description of the methods used to detect violations of the proportional hazards assumption and the construction of the weights used to address that.

Answer: We now provide a more detailed description of the methods used to detect violations of the proportional hazards assumption and the construction of the weights used to address that. We also included in the current submission a link to the statistical code used for these analyses, which has been deposited in a recognized public source code repository (GitHub, https://github.com/mlsrico/haloperidol_and_covid19)

P9: “We conducted six sensitivity analyses. First, we performed propensity score analyses with inverse probability weighting (IPW) [26,27]. The individual propensities for haloperidol prescription were estimated by a multivariable logistic regression model that included as covariates the same variables used in the primary analyses. The predicted probabilities from the propensity-score model were used to calculate the stabilized inverse-probability-weighting weights [26]. Associations between haloperidol use and the two outcomes were then estimated using IPW Cox regression models. In cases of non-balanced covariates, IPW multivariable Cox regression models adjusting for these non-balanced covariates were also performed. Kaplan-Meier curves were performed using the inverse-probability-weighting weights [24,25].”

P10: “For all associations, we performed residual analyses to assess the fit of the data, check assumptions, including proportional hazards assumption using proportional hazards tests and diagnostics based on weighted residuals [29], and examined the potential presence and influence of outliers [30].”

P10: “All analyses were conducted in R software version 2.4.3. Statistical code used for the main analyses has been deposited in a recognized public source code repository (GitHub, https://github.com/mlsrico/haloperidol_and_covid19).”

16/ The analyses are inadequately documented. Methodological statements such as the sentence on line 205 of the manuscript fail to convey any useful or relevant information. The authors should instead provide their code as part of supplemental material.

Answer: Following this comment, we have suppressed this sentence (“To improve the quality of result reporting, we followed the recommendations of The Strengthening the Reporting of Observational Studies in Epidemiology (STROBE) Initiative [25].”) and have included in the current submission a link to the statistical code used for these analyses, which has been deposited in a recognized public source code repository (GitHub, https://github.com/mlsrico/haloperidol_and_covid19)

Following this comment, we have modified the manuscript (please see responses to comment #15).

17/ The authors provide a link to the data on line 362. That link points to a general French-language website rather than to specific data. Most readers will have great difficulty navigating through that website to find the data. I could not. Instead, the data file and metadata should be posted in a publicly accessible archive or provided as supporting information.

Answer: The clinical data that support the findings of this study are available at https://eds.aphp.fr// (only French-language website) with the permission of AP-HP Health Data Warehouse (Entrepôt de Données de Santé (EDS). Restrictions apply to the availability of these data, which were used under license for this study. We now specify this point in the Competing Interests Statement and Data Availability Statement, as also asked by the Editor.

Reviewer #3

18/ The Authors examined the treatment effect of haloperidol use in hospitalized COVID-19 patients at the APHP Greater Paris University hospitals. The primary endpoint was a composite of intubation or death and the secondary endpoint was discharge to home among survivors in time-to-event analyses. To control selection bias due to nonrandom assignment of treatments, the authors built a propensity score model with demographic and clinical factors and medication uses as predictors for haloperidol use. The authors then calculated stabilized weights and applied the weights to a multivariable Cox regression model with the start time being the date of hospital admission. Sensitivity analyses were also conducted including a multivariable Cox regression model with the same variables as in the propensity score model and a univariate Cox regression model in a matched analytic sample. The manuscript was well organized and clearly written. My major concerns are:

Answer: We thank the reviewer for this positive comment.

19/ For those who received the treatment not on admission date but during hospitalization, their follow-up periods consisted of two parts: untreated and treated. The authors did not consider these two different follow-up periods in their analyses. Related to comment#19, immortal bias may arise because patients had to be alive before receiving the treatment compared to the control group who could die any time after hospital admission.

Answer: Following this comment and Reviewer 1 comment #5, we have rerun all statistical analyses while considering a new definition of haloperidol use to only include patients actively taking haloperidol at study baseline, in order to reduce the risk of immortality bias. Haloperidol use is now defined as receiving haloperidol within the first 48 hours of hospital admission and before the end of the index hospitalization or intubation or death. We used this delay because we considered that, in a context of overwhelming of all hospital units during the COVID-19 peak incidence, patients may not have received or been prescribed their usual treatment the first day of their admission, or the treatment may not have been recorded in the computerized medication administration data the first day of admission. We chose to exclude from the main analyses patients who were prescribed haloperidol more than 48 hours from hospital admission or after intubation, in order to reduce the risk of immortality bias, help interpret results, and because adjustment variables were measured at the time of hospital admission. Furthermore, we performed additional sensitivity analyses considering haloperidol use as a time dependent variable, including all participants who received haloperidol during the hospitalization from study baseline until the end of the index hospitalization or intubation or death, in order to account for untreated and treated periods among patients receiving haloperidol. Results were similar, supporting the robustness of our findings.

Following this comment, we redid all Figures and Tables and modified the manuscript (please see responses to comment #5).

20/ It does not seem right to include medications prescribed during hospitalization in propensity score model. This is because these medications could be given after patients received haloperidol treatment.

Answer: We agree with the reviewer. Unfortunately, data on date of prescribing were not available for most psychotropic medications. Because these variables were independently associated with the outcomes, we chose to include them as covariates in the analyses. However, we performed additional sensitivity analyses and examined whether our results were similar when not including psychotropic medications as covariates in the main analyses. Results were similar, supporting the robustness of our findings. 

 Following this comment, we added a table (S6 Table) and modified the manuscript as follows:

P10: “Finally, because psychotropic medications other than haloperidol could have been prescribed to patients after they received haloperidol, we examined whether the results were similar when not including these variables as covariates in the main analyses.”

P18: “Finally, models imputing missing data using multiple imputation yielded very similar results as in the main analyses (S5 Table), as did models not including psychotropic medications other than haloperidol as covariates (S6 Table).”

21/ Table 1 shows that after applying weighting using IPW, several variables were still significantly different between the two groups. I was wondering if the authors explored higher orders and interactions between covariates to improve the propensity score model. Another way is to include these unbalanced covariates in the Cox model.

Answer: Following this comment, we performed as additional sensitivity analyses IPW multivariable Cox regression models adjusting for these non-balanced covariates and modified the manuscript as follows: 

 P9: “First, we performed propensity score analyses with inverse probability weighting (IPW) [26,27]. The individual propensities for haloperidol prescription were estimated by a multivariable logistic regression model that included as covariates the same variables used in the primary analyses. The predicted probabilities from the propensity-score model were used to calculate the stabilized inverse-probability-weighting weights [26]. Associations between haloperidol use and the two outcomes were then estimated using IPW Cox regression models. In cases of non-balanced covariates, IPW multivariable Cox regression models adjusting for these non-balanced covariates were also performed. Kaplan-Meier curves were performed using the inverse-probability-weighting weights [24,25].”

P17-18: “Similar non-significant results were found in the inverse probability weighting analyses adjusting for the two unbalanced covariates (i.e. any antipsychotic medication other than haloperidol and any other psychotropic medication) (i.e., HR, 1.32; 95% CI, 0.65 to 2.71; p=0.441 for the primary outcome; and HR, 1.18; 95% CI, 0.71 to 1.95; p=0.528 for the secondary outcome), as well as in the analyses including all participants who received haloperidol during the hospitalization until the end of the index hospitalization or intubation or death and considering haloperidol use as a time dependent variable (S3 Table).”

22/ In line 173, the authors mentioned ‘weighted Cox regression models were used when proportional hazards were not met’. I am not aware of such approach for handling the nonproportional hazard issue. Please elaborate this approach and cite the reference(s).

Answer: Following this comment we have added the following reference (Dunkler D, Ploner M, Schemper M, Heinze G. Weighted Cox Regression Using the R Package coxphw. J Stat Softw. 2018;84: 1–26. doi:10.18637/jss.v084.i02), included the corresponding statistical code in R (through a link to a recognized public source code repository (GitHub, https://github.com/mlsrico/haloperidol_and_covid19)), and modified the manuscript to clarify this issue. 

P8: “Weighted Cox regression models were used when proportional hazards assumption was not met [23].”

P10: “For all associations, we performed residual analyses to assess the fit of the data, check assumptions, including proportional hazards assumption using proportional hazards tests and diagnostics based on weighted residuals [29], and examined the potential presence and influence of outliers [30].”

23/ I recommend including those who did not consent in Figure 1.

Answer: Patients who did not consent to participate were excluded prior to the construction of the database. Therefore, unfortunately, we were not able to estimate their exact number. Following this comment, we specified this point in the manuscript as follows:

P6: “Participants who did not consent to participate in the study were excluded prior to the construction of the database.”

24/ I am not sure if the linear form of the dosage is appropriate in the Cox model. I was wondering if the authors had tried different functional forms for this variable when studying the dosage effect.

Answer: We agree with the reviewer. Following this comment, we now examine a potential dose-effect relationship by testing the association between the daily dose received (dichotomized at the median value) with the two endpoints among patients who received haloperidol.

P10: “Second, we examined a potential dose-effect relationship by testing the association between the daily dose received (dichotomized at the median value) with the two endpoints among patients who received haloperidol.”

25/ The power estimation did not mean much here because weighting was not considered.

Answer: We agree with the reviewer. However, we followed the recommendations of The Strengthening the Reporting of Observational Studies in Epidemiology (STROBE) Initiative, which recommends the report of power estimation. Nonetheless, we agree that this estimation has limitations that we now better recognize.

P18: “A post-hoc analysis indicated that in the full sample, we had 80% power to detect unweighted and unadjusted hazard ratios of at least 0.20 / 2.41 for the primary endpoint and 0.27 / 5.57 for the secondary endpoint, while we had 80% power to detect unweighted and unadjusted hazard ratios of at least 0.39 / 2.18 for the primary endpoint and 0.39 / 3.07 for the secondary endpoint in the matched analytic sample.”

P19-20: “Second, given the limited number of patients who received haloperidol, our analyses were powered to detect only substantial effect sizes (i.e., 80% power to detect unweighted and unadjusted hazard ratios of at least 0.20 / 2.41 for the primary endpoint and 0.27 / 5.57 for the secondary endpoint in the full sample). In addition, the fact that the inverse probability weighting analyses in the full sample were not successful in balancing two covariates between the two groups (i.e. exposed to haloperidol vs. non-exposed) likely has led to reduced statistical power.”

We warmly thank the Reviewers for their strong help in improving our manuscript.

---

## [Decision Letter · Decision Letter 1]

18 Jan 2021

PONE-D-20-30929R1

Observational Study of Haloperidol in Hospitalized Patients with COVID-19

PLOS ONE

Dear Dr. Marina Sánchez,

Thank you for submitting your manuscript to PLOS ONE. After careful consideration, we feel that it has merit but does not fully meet PLOS ONE’s publication criteria as it currently stands. Therefore, we invite you to submit a revised version of the manuscript that addresses the points raised during the review process.

ACADEMIC EDITOR: The reviewers have still raised several points which we believe additional modifications are necessary to improve the manuscript, taking into account the reviewers' remarks. Please consider and address each of the comments raised by the reviewers.

We look forward to receiving your revised manuscript.

Kind regards,

Wisit Cheungpasitporn, MD

Academic Editor

PLOS ONE

Reviewers' comments:

Reviewer's Responses to Questions

**Comments to the Author**

1. If the authors have adequately addressed your comments raised in a previous round of review and you feel that this manuscript is now acceptable for publication, you may indicate that here to bypass the “Comments to the Author” section, enter your conflict of interest statement in the “Confidential to Editor” section, and submit your "Accept" recommendation.

Reviewer #1: All comments have been addressed

Reviewer #2: (No Response)

Reviewer #3: (No Response)

2. Is the manuscript technically sound, and do the data support the conclusions?

Reviewer #1: Yes

Reviewer #2: Yes

Reviewer #3: Yes

3. Has the statistical analysis been performed appropriately and rigorously? 

Reviewer #1: Yes

Reviewer #2: Yes

Reviewer #3: Yes

4. Have the authors made all data underlying the findings in their manuscript fully available?

Reviewer #1: Yes

Reviewer #2: Yes

Reviewer #3: Yes

5. Is the manuscript presented in an intelligible fashion and written in standard English?

Reviewer #1: Yes

Reviewer #2: Yes

Reviewer #3: Yes

6. Review Comments to the Author

Reviewer #1: The authors have replied in a very detailed and successful way to previous comments. The revised version of the article is much improved.

---I noted some inconsistent data reported in TABLE 1. The authors have revised their article so that the matched analysis is now the primary analysis. As such, they have stated that the matching was successful in balancing the exposed haloperidol persons (N=39) and referent matched non-haloperidol persons (N=156). There appears to be some issue with the reported balancing in TABLE 1 between these groups (exposed versus matched unexposed). The reported matching on age and sex seem reasonable. However, the reported matching seems to be flipped or incorrectly reported in TABLE 1 for smoking yes (28.2% vs. 71.8%) for obesity yes (23.1% vs. 76.9%) and for the medical conditions and other medications used (the last 5 characteristics in TABLE 1). Please ensure that there has not be some accidental number transposition for these characteristics in column 1 (exposed haloperidol N=39) and column 3 (non-exposed matched N=156).

Reviewer #2: Please soften the conclusions stated on lines 69 and 387. The data and analyses support only the weaker conclusion that no association was detected.

Reviewer #3: The authors addressed all the reviewers concerns and did a great job. I only have a few minor suggestions.

In the result sections

1. Characteristics of the cohort. I would suggest replacing the mean by median and delete the mean and std for those variables. Since the distributions of those variables are skewed, the mean and std does not provide much useful info.

2. Study Endpoints

For the secondary endpoint, for some reason the authors reported the results from the ‘IPW’ instead of the matched results as the primary analysis. I think it should be the results from the match sample.

7. PLOS authors have the option to publish the peer review history of their article (what does this mean?). If published, this will include your full peer review and any attached files.

Reviewer #1: **Yes: **Brandon R. Grossardt, MS

Reviewer #2: No

Reviewer #3: No

---

## [Author Response · Author response to Decision Letter 1]

18 Jan 2021

Editor

1/ The reviewers have still raised several points which we believe additional modifications are necessary to improve the manuscript, taking into account the reviewers' remarks. Please consider and address each of the comments raised by the reviewers.

Answer: We thank the reviewers for these additional helpful comments, which have allowed us to strengthen the manuscript. A point-by-point response is included below, and the corresponding changes have been made in the manuscript and have been highlighted in yellow in a separate file.

Reviewer #1

2/ The authors have replied in a very detailed and successful way to previous comments. The revised version of the article is much improved.

Answer: We thank the Reviewer for this positive comment.

3/ I noted some inconsistent data reported in TABLE 1. The authors have revised their article so that the matched analysis is now the primary analysis. As such, they have stated that the matching was successful in balancing the exposed haloperidol persons (N=39) and referent matched non-haloperidol persons (N=156). There appears to be some issue with the reported balancing in TABLE 1 between these groups (exposed versus matched unexposed). The reported matching on age and sex seem reasonable. However, the reported matching seems to be flipped or incorrectly reported in TABLE 1 for smoking yes (28.2% vs. 71.8%) for obesity yes (23.1% vs. 76.9%) and for the medical conditions and other medications used (the last 5 characteristics in TABLE 1). Please ensure that there has not be some accidental number transposition for these characteristics in column 1 (exposed haloperidol N=39) and column 3 (non-exposed matched N=156).

Answer: We warmly thank the Reviewer for noticing this accidental number transposition, which is now corrected (see Table 1).

Reviewer #2

4/ Please soften the conclusions stated on lines 69 and 387. The data and analyses support only the weaker conclusion that no association was detected.

Answer: Following this comment, we softened these two conclusion sentences as follows:

P4, L68-71: “Findings from this multicenter observational study suggest that haloperidol use prescribed at a mean dose of 4.5 mg per day (SD=5.2) for a mean duration of 8.4 days (SD=7.2) may not be associated with risk of intubation or death, or with time to discharge home, among adult patients hospitalized for COVID-19.”

P20, L386-388: “Findings form this multicenter observational study suggest that haloperidol prescribed at a mean daily dose of 4.5 mg per day (SD=5.2) for a mean duration of 8.4 days (SD=7.2) may not be associated with risk of intubation or death, or with time to discharge home, among patients hospitalized for COVID-19.”

Reviewer #3

5/ The authors addressed all the reviewers concerns and did a great job. I only have a few minor suggestions.

Answer: We thank the Reviewer for this positive comment.

6/ In the result sections

1. Characteristics of the cohort. I would suggest replacing the mean by median and delete the mean and std for those variables. Since the distributions of those variables are skewed, the mean and std does not provide much useful info.

Answer: Following this comment, we now present the median values as the main results in this paragraph. However, because we agree with the Reviewer that the distributions of those variables are skewed, we let the information into brackets about the distribution of these variables, i.e., SD, mean, 1st quartile, and 3rd quartile, for interested readers. 

Following this comment, we corrected the manuscript as follows:

P10-11: “Of the 17,076 hospitalized adult patients with a positive COVID-19 RT-PCR test, 1,908 patients (11.2%) were excluded because of missing data or their young age (i.e. less than 18 years of age). Of the 86 adult patients who received haloperidol at any time during the visit, 47 (54.7%) patients were excluded because they received it more than 48 hours from hospital admission (N=47, 54.7%) or after intubation (N=16, 18.6%). Of the remaining 15,121 inpatients, 39 patients (0.3%) received haloperidol at baseline (i.e., within the first 48 hours of hospital admission) at a median daily dose of 3.0 mg per day (SD=5.2; mean=4.5; first quartile=1.0; third quartile=5.0; range=0.5-20.0 mg) for a median duration of 7.0 days (SD=7.2; mean=8.4; first quartile=2.5; third quartile=12.0; range=1-26). Of these 39 patients, 4 patients (10.2%) had a medication administration by intramuscular injection. Median delay between study baseline and haloperidol initiation was lower than 0 day (SD=1.0; mean=0.6; first quartile=0.0; third quartile=1.0; range=0-2 days) (S1 Figure).

First positive COVID-19 RT-PCR tests were obtained after a median delay of 1.2 days (SD=12.7) from study baseline. This delay was significantly but not substantially different between patients receiving or not receiving haloperidol [median in the exposed group=1.0 day (SD=11.2); median in the non-exposed group=1.2 days (SD=12.8); Mood’s median test Chi-square=3.76, p=0.001].

 Over a median follow-up of 7 days (SD=17.9; mean=13.8; first quartile=1.0; third quartile=22.0; range: 1-98 days), 2,024 patients (13.4%) had a primary end-point event and 10,179 patients (77.6%) were discharged home at the time of study end on May 1st. Patients receiving haloperidol had a median follow-up of 7 days (SD=14.0; mean=10.9; first quartile=4; third quartile=22; range: 1-80 days), while the non-exposed group had a median follow-up of 7 days (SD=17.9; median=13.8; first quartile=1; third quartile=22; range: 1-98 days) [Mood’s median test Chi-square=0.41, p=0.679].”

7/ 2. Study Endpoints

For the secondary endpoint, for some reason the authors reported the results from the ‘IPW’ instead of the matched results as the primary analysis. I think it should be the results from the match sample.

Answer: We warmly thank the Reviewer for noticing this typo, which is now corrected.

P15, L304-307: “Haloperidol use was significantly and negatively associated with the secondary endpoint in the crude, unadjusted analysis (HR, 0.24; 95% CI, 0.13 to 0.44; p<0.001), but this association was not significant in the primary analysis using a univariate Cox regression model in the matched analytic sample (HR, 1.30; 95% CI, 0.74 to 2.28; p=0.355) (Fig 2; Table 2).”

We warmly thank the Reviewers for their strong help in improving our manuscript.

---

## [Decision Letter · Decision Letter 2]

2 Feb 2021

Observational Study of Haloperidol in Hospitalized Patients with COVID-19

PONE-D-20-30929R2

Dear Dr. Marina Sánchez,

We’re pleased to inform you that your manuscript has been judged scientifically suitable for publication and will be formally accepted for publication once it meets all outstanding technical requirements.

Kind regards,

Wisit Cheungpasitporn, MD

Academic Editor

PLOS ONE

Additional Editor Comments:

I reviewed the revised manuscript and the response to minor reviewers' comments. Revised Manuscript is well written. All comments have been addressed and thus accepted for publication.

Reviewers' comments:

Reviewer's Responses to Questions

**Comments to the Author**

1. If the authors have adequately addressed your comments raised in a previous round of review and you feel that this manuscript is now acceptable for publication, you may indicate that here to bypass the “Comments to the Author” section, enter your conflict of interest statement in the “Confidential to Editor” section, and submit your "Accept" recommendation.

Reviewer #1: All comments have been addressed

Reviewer #2: All comments have been addressed

Reviewer #3: All comments have been addressed

2. Is the manuscript technically sound, and do the data support the conclusions?

Reviewer #1: Yes

Reviewer #2: Yes

Reviewer #3: Yes

3. Has the statistical analysis been performed appropriately and rigorously? 

Reviewer #1: Yes

Reviewer #2: Yes

Reviewer #3: Yes

4. Have the authors made all data underlying the findings in their manuscript fully available?

Reviewer #1: Yes

Reviewer #2: Yes

Reviewer #3: Yes

5. Is the manuscript presented in an intelligible fashion and written in standard English?

Reviewer #1: Yes

Reviewer #2: Yes

Reviewer #3: Yes

6. Review Comments to the Author

Reviewer #1: (No Response)

Reviewer #2: (No Response)

Reviewer #3: The author did a good job addressing my concerns. I am satisfied with the revision and have no further questions.

7. PLOS authors have the option to publish the peer review history of their article (what does this mean?). If published, this will include your full peer review and any attached files.

Reviewer #1: **Yes: **Brandon R. Grossardt, MS

Reviewer #2: No

Reviewer #3: No

---

## [Editor Report · Acceptance letter]

12 Feb 2021

PONE-D-20-30929R2 

Observational Study of Haloperidol in Hospitalized Patients with COVID-19 

Dear Dr. Sánchez-Rico:

I'm pleased to inform you that your manuscript has been deemed suitable for publication in PLOS ONE. Congratulations! Your manuscript is now with our production department. 

Kind regards, 

on behalf of

Dr. Wisit Cheungpasitporn 

Academic Editor

PLOS ONE